# Development of a Molecular Aptamer Beacon Applied to Magnetic-Assisted RNA Extraction for Detection of Dengue and Zika Viruses Using Clinical Samples

**DOI:** 10.3390/ijms232213866

**Published:** 2022-11-10

**Authors:** Amanda Gabrielle da Silva, Luiz Ricardo Goulart, Philipp Löffler, Christian Code, Adriana Freitas Neves

**Affiliations:** 1Institute of Physics, Postgraduate Program in Exact and Technological Sciences, Universidade Federal de Catalão, Catalão 75704-020, Brazil; 2Nanobiotechnology Laboratory, Institute of Biotechnology, Universidade Federal de Uberlândia, Uberlândia 38402-022, Brazil; 3Department of Physics, Chemistry and Pharmacy, University of Southern Denmark, Campusvej 55, 5230 Odense, Denmark; 4Dianox ApS, Fruebjergvej 3, 2100 København, Denmark; 5PhyLife Physical Life Sciences, Department of Physics, Chemistry and Pharmacy, University of Southern Denmark, Campusvej 55, 5230 Odense, Denmark; 6Institute of Biotechnology, Molecular Biology Laboratory, Universidade Federal de Catalão, Catalão 75704-020, Brazil

**Keywords:** flavivirus, oligonucleotides, hybridization, diagnosis, RT-PCR

## Abstract

Limitations in the detection of cocirculating flaviviruses such as Dengue and Zika lead us to propose the use of aptameric capture of the viral RNA in combination with RT-PCR (APTA-RT-PCR). Aptamers were obtained via SELEX and next-generation sequencing, followed by colorimetric and fluorescent characterizations. An APTA-RT-PCR assay was developed, optimized, and tested against the viral RNAs in 108 serum samples. After selection, sequence APTAZC10 was designed as a bifunctional molecular beacon (APTAZC10-MB), exhibiting affinity for the viral targets. APTA-RT-PCR was able to detect Dengue and Zika RNA in 43% and 8% of samples, respectively. Our results indicate that APTAZC10-MB and APTA-RT-PCR will be useful to improve the detection of Dengue and Zika viruses in a fast molecular assay for the improvement of infectious disease surveillance.

## 1. Introduction

Zika virus (ZIKV), an arbovirus belonging to the Flaviviridae family, was initially isolated from rhesus monkeys in the Zika forest of Uganda in 1947 [1]. Its transmission cycle involves mainly vectors of the genus Aedes [2]. From 1947 to 2007, ZIKV was geographically restricted to African and Southeast Asian regions. The first epidemic was reported in The Island of Yap and in the Federated States of Micronesia. Other outbreaks have since been reported across the Pacific Ocean region, including one in French Polynesia in 2013–2014, with thousands of confirmed cases in which individuals manifested neurological complications associated with Guillain-Barré syndrome [3,4]. In early 2015, the first case of Zika in Brazil was confirmed, which quickly spread throughout the country due to the great mobility of its population and the ubiquity of the transmitting vectors [5]. Sexual transmission [6] and the association of microcephaly with ZIKV [7]—detected either in the placenta or in the blood of microcephalic newborns [8,9]—highlighted the importance of seeking better diagnostic and therapeutic tools. ZIKV causes symptoms similar to other flavivirus infections, such as Dengue virus (DENV), and its differential diagnosis plays a decisive role in obtaining fast reliable results, requiring both high sensitivity and specificity. Because of the similarity in the viruses, the induced antibodies may exhibit cross-reactivity. This presents a problem in diagnosis, especially in countries where different arboviruses co-circulate [10,11]. The diagnosis of arboviruses is a challenge, since the commonly used serological method, ELISA (Enzyme-Linked Immunosorbent Assay), is based on the interaction between antigens and antibodies, and can result in a low specificity when differentiating between different flaviviruses infections [12]. As for Severe acute respiratory syndrome coronavirus 2 (SARS-CoV-2) [13], the gold standard for diagnosis has been the detection of viral RNA by reverse transcription followed by the polymerase chain reaction (RT-PCR). However, negative results of ZIKV infection cannot be excluded due to the small number of viral particles present during the sample collection in the early stages of the disease [14]. Furthermore, co-infections are not considered in the clinical management in the Brazilian National guidelines [15,16], which present ways of classifying symptoms, proposals for diagnosis, and the treatment of Dengue or Zika.

Antibody-based detection methods are widely used but have a high production cost and a limited specificity of detection. Nucleic acid aptamers have emerged as new tools in viral detection antiviral agents due to their ability to bind with high affinity for different biological macromolecules. The interaction between the aptamer and its target is provided by the well-defined three-dimensional conformation of the aptamer, which forms the basis of its great potential for use in the detection and treatment of viruses [17,18]. Recent literature has demonstrated the potential application of aptamers that bind to different targets in association with diseases such as Dengue fever [19,20,21,22,23]. Aptamers are single-stranded RNA or DNA oligonucleotides that have a high affinity towards a molecular target. They are isolated by the SELEX method (Systematic Evolution of Ligands by Exponential Enrichment), a process involving a random library of oligonucleotides subjected to successive rounds of binding to an immobilized target, followed by separation and amplification of binding sequences to enrich the ligand library [24]. We previously selected aptamers that would target the 5′ untranslated region (UTR) of the Dengue virus genome by using Sanger sequencing [21] and have used the SELEX method to select aptamers binding in 5′-UTR of ZIKV analyzed by next-generation sequencing (NGS). This region was chosen due to the recent evidence that the 5′ and 3′ non-coding RNAs can induce the infectious process, playing a critical role in regulating the virus–host interaction during infection [25]. The conformational structures found in the 5′ region of the genomic RNA appear to function as promoters for the action of the RNA-dependent RNA polymerase (RdRp) [26]. The 3′UTR region also plays an important role in the infectious process; however, it has been reported that observed differences in size of the 3′UTR-DENV sequences were correlated with sequences from different geographic locations. The size variability was detected due to deletions, mutations, and imperfect repeats in domain I of the region in strains [27]. Furthermore, the secondary RNA structures present in the 3′-UTR-ZIKV are less conserved compared to other flaviviruses; changes in the first loop conformation in 3′-UTR-ZIKV of a post-epidemic strain were detected due to variants with nucleotide substitutions [28]. 

Herein, we developed an aptamer, denominated APTZC10-MB, in a methodology for detecting ZIKV and DENV by combining the sensitivity of molecular RT-PCR techniques with the affinity of the aptamer in the pre-purification of viral RNA directly in the serum sample. The goal was to select a bifunctional aptamer that captures the viral RNA, and then submits it to RT-PCR that will specifically amplify the target, thus reducing sample processing and analysis costs related to RNA extraction. The APTA-RT-PCR was able to detect viral RNA in infected serum samples showing the ability of this aptamer in future applications of technologies in biomedical research, especially in a rapid and low-cost method of genetic material detection.

## 2. Results

### 2.1. Target Production and SELEX

The workflow of the SELEX method used in this study to produce ligands against 5′UTR from the ZIKV genome is illustrated in Figure 1. The consensus region obtained from the alignment of the sequences deposited in the National Center for Biotechnology Information (NCBI) database corresponds to positions 15 to 109 in 5′UTR-ZIKV and, as expected, the amplified product (Figure 1A) presented approximately 95 bp. The purified amplicons were used as a template for the asymmetric PCR, in which only the biotinylated forward primer was used, to ensure biotinylation of the 5′-end of the target for binding to streptavidin-coated paramagnetic particles (Figure 1B). In addition, by using dNTPs containing dUTP, we generated RNA-like analogs of the 5′UTR-ZIKV. Electrophoresis analysis using 1% agarose gel showed that the ligands obtained from the eighth round ranged from 50 to 250 nucleotides in length (Figure 1C). The enrichment of the oligonucleotide library was verified by both the quantification of the pool of ligands obtained during the rounds and the dot-blot hybridization assay. The quantification results showed that the library complexity decreased during the rounds, which is expected given the elimination of non-binding sequences, which occurred mostly in the second round (Figure 1D). In addition, a decrease in complexity was also observed in the fifth round, which corresponds to an increased stringency, where binders with lower affinity for the target were removed. The dot-blot hybridization assay allowed evaluation of the enrichment and the affinity of the aptamer pool with the target. The precipitates observed on the membrane are products of a redox reaction, where the alkaline phosphatase cleaved the phosphate group present on the NBT/BCIP substrate, producing diformazan precipitate, thus indicating the interaction of the samples with the biotinylated probe (Figure 1E). Among the pools of ligands fixed on the membrane, staining was observed with more intensity in the eighth round (4th dot), thus showing affinity of this pool with the target and enrichment of the ligands in the selection process.

### 2.2. Analysis of the Sequences from Next-Generation Sequencing

In total, the sequencing of ligands for the 5′UTR-ZIKV from the last round resulted in 4,169,589 oligonucleotide sequences. The length of these ligands varied from 29 to 462 nucleotides with different frequencies and presented an average size of 88-bp (Figure 2A). The raw data obtained were processed and the sequences whose length was less than 50-bp and more than 250-bp were eliminated, and the remaining aptamers presenting the variable and fixed sequences were selected.

The homologous sequences were clustered, where the ligands with higher frequencies are expected to be the most promising aptamers. The data were organized into 156,180 clusters, which were ordered from the most frequent to the least frequent sequences of ligands. The representative sequences of the clusters from 1 to 30 were analyzed. It was observed that in the sequence alignment, the first five clusters could be classified into three different groups, according to the linear motifs found: “ATGGG”, “TGGGAT” and “AGGG”. These motifs are variations of “T/AGGG”, which was observed more frequently in the sequences. The APTAZC10 representing Cluster 1 was selected for the characterization tests. It is important to note that APTAZC10 presented the highest frequency (Figure 2B) as a ZIKV ligand and was also one of the sequences among previously obtained DENV ligands in our lab [21]. In this sense, the APTAZC10 is indicated to be one of the bifunctional molecules as a ligand for DENV and ZIKV by using the capture process with paramagnetic particles followed by real-time PCR (Appendix A). Analyzing the cycle threshold (Ct), the APTZC10 showed the lowest cycle threshold among the aptamers tested (Appendix A). The melting curves confirmed the amplification of the captured molecules by comparing the Tm of the negative control (Appendix A). Furthermore, in this assay, it was possible to identify a specific aptamer against DENV (Appendix A), and together these results corroborated data from NGS, indicating APTZC10 aptamer as a potential ligand for both flaviviruses.

### 2.3. The Characterization of APTAZC10-MB

Based on the prediction of the secondary structure of APTAZC10, we designed a molecular beacon for this aptamer, denominated APTAZC10-MB, with base changes in the stem and without the fixed 5′ and 3′ sequences. The insertion of five bases at both ends induced the stem-loop conformation as a characteristic of a molecular beacon, and together with the dot-blot, the data demonstrated an effective approach in the design of the APTAZC10-MB molecule (Figure 3). The dot-blot for the APTAZC10-MB trial showed no interaction with the targets presented as single strands. 

For both DENV and ZIKV, it was possible to verify the hybridization signal presented in the sequences of the targets amplified, dsDNA 5′UTR-DENV2 and dsDNA 5′UTR-ZIKV. On the other hand, the interaction observed between ssDNA of the 5′UTR-DENV2 and aptamers was less effective when compared with the dsDNA hybridization. We hypothesize that the aptamers will bind with higher affinity and stability to double-stranded structures, mainly for ZIKV, but interactions with DENV ssDNA or dsDNA are possible.

For the 5′UTR-ZIKV as the target, titration varying from 0 to 200 nM demonstrated that increasing concentrations of the target led to an increase in the intensity of TAMRA. This behavior in intensity was not expressive, which can be explained by the influence of the ions present in the buffer (Appendix A) and the thermal stability of the aptamer up to about 55 °C (Appendix A). No significant changes were observed on the APTAZC10-MB structure in the presence of the targets, as verified by the emission of acceptor fluorescence. The calculated K_d_ was 2.62 nM (Figure 4A) for DENV and 25.85 nM for ZIKV (Figure 4B), indicating that APTAZC10-MB presented a better interaction with DENV, as observed especially for APTAZC10 in the dot-blot.

### 2.4. APTA-RT-PCR Development for DENV and ZIKV Detections

The samples used to test the APTA-RT-PCR for DENV and ZIKV detections are represented in Figure 5. In the age range from 35 to 39 years, 57% of the population was composed of males, whereas females were more prevalent in the age groups from 25 to 44 years and over 64 years (Figure 5A). Considering serological data as a routine trial, 55 samples were negative and 53 positives for the Dengue virus; among these, 41% were reactive to IgG and IgM, 25% for IgM or IgG, and 30% of the cases were reactive to NS1 antibodies (Figure 5B). No cases of ZIKV were previously registered. Reported, herein, are the results for an optimized assay with 108 samples in total, of which 46 cases tested positive and 62 negatives for Dengue. Positive cases for ZIKV were detected in 6.5% of the samples, and two of the positive detections were coinfections (1.9%) (Figure 5C). Five samples analyzed in a 24-well plate in C6/36 cell culture (methods not shown) confirmed the detection by APTA-RT-PCR due to the cytopathic effect observed (Appendix A). After RNA extraction of the culture cells, RT-PCR was again performed, and a profile of the expected band was observed (Appendix A). Although the RNA extraction and RT-PCR is a gold standard for detecting both flaviviruses [29], for a comparative analysis our data were evaluated against serological tests. Thus, the diagnostic parameters such as sensitivity and specificity in the comparative analysis indicated APTA-RT-PCR sensitivity ranging from 35% to 60% and specificity between 52% and 63% when compared to blood count or serological-based tests (Appendix A). The extraction and subsequent RT-PCR assays were just performed as an initial pilot for our APTA-RT-PCR assay development (methods not shown), so it was not possible to correlate RT-PCR with and without aptamer use. No significant correlation (*p* < 0.05) was found among tests based on serology or blood count with the APTA-RT-PCR (Appendix A).

## 3. Discussion

In our study, we described the use of the SELEX method to isolate aptamers that are ligands of the 5′UTR region of the ZIKV viral genome and their use in serum sample detection. The diverse characteristics presented by these ligands allow their application in several areas of research [30], aiming at improving the technologies employed. These molecules have become potential tools for the development of cheaper and faster development of diagnostic and therapeutic platforms, with this being the focus of several researchers on ligand selection against viral diseases [18]. Recent studies have reported the use of oligonucleotide aptamers to detect ZIKV. Lee and Zeng [31] used an ELISA assay composed of ssDNA aptamers binding to the NS1 protein of the ZIKV genome. Saraf and collaborators [32] developed a microfluidic device using aptamers and gold nanoparticles (AuNP) to detect both Zika and Chikungunya viral envelope proteins. Combining peptide aptamers and bioinformatics, Kim et al. [33] used molecular modeling and experimental testing to study the interaction between peptides and viruses in serum and urine samples and to select the molecule with greater affinity. 

As done by Oteng, Gu, and McKeague [34], we selected the aptamer that presented the highest frequency in the results obtained from NGS that corroborated the data of the dot-blot techniques and real-time PCR, which showed APTZC10 and/or APTAZC10-MB as potential ligand(s). Although the dot-blot hybridization assay has a qualitative character, it can be quite an effective diagnostic tool since it is perceptible to the naked eye; furthermore, the literature notes that the dot-blot is employed to verify the enrichment of ligands and their affinity toward the target [35,36,37]. Several advantages are gained from utilizing NGS to obtain ligand sequences as performed for our aptamers: 1. It has the potential to completely replace Sanger cloning steps; 2. It does not limit the number of aptamers to be sequenced; 3. It allows the evaluation of the enrichment during the process of selection and identification of sequences with low frequency [38]. NGS provides information on millions of sequences and can be more informative, although with a lower resolution of the set of ligands, whereas, Sanger sequencing can be used for aptamer selection [39]. The best ligands were found in the first selection cycle with the use of NGS, and their enrichment became evident in the second and third rounds [40]. Aptamers selected with the use of NGS presented greater affinity and specificity compared to those obtained with conventional cloning and sequencing methods [41].

The analysis of aptamers using in silico methods revealed important data about the binding region. The prediction of the structural conformation can be made using a combination of free online tools such as Mfold, Nupack and RNAstructure [42,43,44], which can analyze the sequences by minimizing free energy. Furthermore, by predicting the aptamer structure, it is possible to propose changes in the bases to promote modifications in the conformation of the aptamer aiming at future applications for the selected ligand. The in silico predictions made in other studies [45,46,47] were confirmed in our experimental assays, suggesting that they were quite reliable and can be used for future applications. Our studies aimed to design a molecular aptamer beacon (MB) due to the possibilities presented by this molecule, combining the high affinity of aptamers and the sensitivity of the MB’s transduction signal for detecting DNA and RNA sequences [48]. The APTZC10-MB projected with the removal of the fixed sequences and the insertion of the bases was performed according to Zheng et al. [49], such as the GC content and the length of the stem that could not exceed the size of five to seven bases.

The methods used to diagnose Zika, Dengue, and other arboviruses can vary in sensitivity and specificity [50] (Table 1). In Brazil, the laboratory methods commonly used are serological tests for detecting IgM or IgG antibodies and the NS1 viral protein; viral isolation; RT-PCR; and immunohistochemistry. The method is chosen according to the number of days the patient presents symptoms of infection [15] and the Dengue disease severity. NS1 antigen is detected in the first five days of infection [50] and is recommended for the diagnosis of a patient with acute Dengue infection, but not as a routine test for asymptomatic patients [51]. RT-PCR tests are recommended for epidemiological monitoring, due to their sensitivity, in addition to the possibility of using different biological samples for detection (blood, serum, urine) and of identifying coinfections [50]. Postmortem diagnosis is used for epidemiological surveillance [51,52].

Among such methods, the most widely used are immunoenzymatic assays for the detection of IgM and IgG antibodies because, in comparison to the others, it has the lowest cost and is easier to perform due to the automation of the method [64]. The cross-reaction in antibody responses to DENV and ZIKV has been reported in many studies in the literature, showing much evidence of cross-reactivity for Dengue IgM ELISA, but not for Dengue ELISA NS1 tests [65]. Van Meer et al. [66] indicated that the NS1 Dengue test was specific; however, a high percentage of cross-reactivity was observed in the IgM and IgG results. Lima et al. [67] reported a specificity of about 99% in the NS1 Dengue test with 92% accuracy in not detecting acute Zika infections. In the results of Tsai et al. [68], the combined diagnosis of ZIKV-NS1 IgG was cross-reactive with samples of secondary Dengue infections. Zaidi et al. [69] observed the cross-reaction of IgM and IgG in patients with ZIKV and DENV infections when testing commercial and in-house ELISA assays. The authors did not observe the specificity of the test with NS1, which also showed a cross-response to both flaviviruses.

On the other hand, the sensitivity and specificity presented by PCR in the detection and quantification of molecules can be combined with the selectivity of aptamers, leading to a methodology called Apta-PCR [70]. APTA-PCR employs two functions where the aptamer acts as a selective binding molecule (biorecognition element) for the target and as a template for the PCR [38]. Therefore, this technique was used to characterize the affinity of post-SELEX aptamers (Appendix A). 

APTA-RT-PCR presented sensitivity of 35% to 60% in comparison with serological results, which can indicate some association between the levels of antibodies and the presence of viral RNA. During the viremia period, frequently between 24 and 48 h before the beginning of symptoms and which can last for 7 days, viral RNA and NS1 protein can be detected in different biological samples, such as blood, serum, and plasma [71]. Viremia levels can be affected by the presence of IgM, which controls viral infections through complement fixation or uptake by phagocytic cells. Viremia reduction occurs due to the clearance of the virus by IgM antibodies. So, one of the disadvantages of the diagnostic technique used in this work is that the capture of viral RNA depends on the viremia level of the patients [72].

In a study by Laue, Emmerich, and Schmitz [73], DENV RNA was not detected in all cases, and the authors pointed out that due to the late collection of the samples, patients had produced anti-dengue antibodies. Since it was not possible to obtain data on the symptoms of the cases used in our study, we can only suggest that the individuals may have undergone laboratory tests during the acute phase of the disease, when viremia levels decrease, and IgM and IgG antibodies levels increase [73].

This study has some limitations. First, although we did not have results from the gold standard techniques, we used the laboratory results that were obtained through enzyme immunoassay and are often used as routine exams of patients with these viral infections due to their low cost and ease of handling [64]. For the WHO [56], a case of Dengue is confirmed by: isolation of the virus in cell culture; or IgM or IgG detected by ELISA between paired samples; or 4-fold increase in IgG titer detected between paired samples; or positive RT-PCR. The CDC [74] defines as a confirmed case: isolation of the virus in cell culture; or IgM detected by ELISA between paired samples; or 4-fold increase in IgG titer detected between paired samples; or positive RT-PCR; or 4-fold increase in Plaque Reduction Neutralization Tests (PRNT) titer between paired samples. In Brazil [15], confirmed cases must present clinical-epidemiological criteria with specific laboratory confirmation: isolation of the virus in cell culture; or positive RT-PCR; reagent IgM ELISA; or positive NS1 antigen screenings; or positive viral antigen by immunohistochemistry.

Thus, despite not constituting a good comparison, the statistical analysis provided us with preliminary data indicating that the technique described in our study has potential but there is a long way to go before we can understand the contribution that this technique can bring to the routine diagnosis of Dengue and Zika virus infections. Second, there are commercial kits of paramagnetic particles that promote RNA extraction. In our assay, the presence in the aptamer gives this test selectivity and specificity in the process of capturing viral RNA. Sulka and colleagues [75] used paramagnetic particles and a molecular beacon to capture the viral RNA and detect it by fluorescence. Third, we cannot make a comparison on relative sensitivity, specificity convenience, and cost with commonly used diagnostic techniques (Table 1). 

However, it was speculated that the Apta-RT-PCR would not be as low in cost as the rapid tests nor provide high sensitivity such as real-time PCR and PRNT. However, compared with the conventional PCR, the advantage of the technique described is that the extraction step would be faster and easier to perform, and furthermore, no hazardous reagents would be used. The specificity and sensitivity can be improved and would be evaluated by gold standard techniques in future projects.

The ability of APTA-RT-PCR to detect the viral genome indicates that this methodology is useful in diagnosing infectious diseases. Additionally, it has the advantage of being specific in viral detection, faster than conventional molecular biology techniques using RNA extraction, and able to identify both Dengue and Zika virus RNA due to the aptamer, which may prove to be a useful tool in differential detection or co-infections. This is not possible just with reactivity of IgM or IgG antibodies, which present cross-reaction with flaviviruses. Future experiments will be carried out to evaluate the sensitivity and ability to distinguish stoichiometric values of aptamers with the number of viral particles. In addition, strategies that will be evaluated to improve the effectiveness of the diagnostic technique include: changes in capture procedures; modification in the sequence and structure of the aptamer [76]; the conjugation of selected aptamers, since there are studies that conjugated aptamers increased target-aptamer affinity [77]; and amplification reaction of other virus coding sequences. Songjaeng et al. [78] reported that amplification of the two different DENV regions, namely the 3′UTR-DENV region and a coding region, increased the diagnostic sensitivity in samples collected during the viremia period.

In the current work, we have presented an aptamer to improve molecular detection of RNA viruses such as Zika and Dengue from a small amount of the serum sample from an infected patient, without the use of RNA extraction. This method reduced both the detection time and the test cost. Furthermore, it has the advantage of being specific in viral detection, faster than conventional molecular biology techniques using RNA extraction, and able to identify both Dengue and Zika virus RNA due to the aptamer, which may prove to be a useful tool in differential detection or co-infections. This is not possible by evaluating only the reactivity of IgM or IgG antibodies, which present cross-reaction for flaviviruses.

## 4. Materials and Methods

### 4.1. Chemicals

The nylon membrane and the AlkPhos Direct Labeling Reagent kit were obtained from GE Healthcare Life Science (São Paulo, Brazil). Oligonucleotides with or without modifications were purchased from GenOne Biotechnologies (Rio de Janeiro, Brazil). Deoxyribonucleotides (dNTPs) were obtained from Ludwig Biotecnologia Ltd.a (Alvorada, Brazil). The 5-Bromo-4-chloro-3-indolyl phosphate (BCIP) and nitro-blue tetrazolium chloride (NBT) were purchased from Sigma-Aldrich (São Paulo, Brazil). Streptavidin MagneSphere^®^ Paramagnetic Particles (PMPs) and T7 RiboMAX™ were obtained from Promega (Madison, WI, USA). Platinum Taq DNA polymerase was purchased from Thermo Fisher Scientific (São Paulo, Brazil). The 5× HOT FIREPol^®^ Eva Green^®^ qPCR Mix Plus (ROX) and M-MLV Reverse Transcriptase RNase H were purchased from Solis BioDyne (São Paulo, Brazil).

### 4.2. 5′UTR-ZIKV as a Target for SELEX

The synthetic sequence of 5′UTR-ZIKV was obtained by aligning the sequences (KU509998.3, KU321639.1, KU926310.1, KU926309.1, KU922923.1, KX051563.1, KU940228.1, NC_012532.1, and KU922960.1), which had been previously deposited and available in the nucleotide database of the NCBI (www.ncbi.nlm.nih.gov, accessed on 22 June 2016). The alignment was made using Clustal Omega [79] and the primers were designed on the software Primer Express ^®^, Version 3.0 (Applied Biosystems, São Paulo, Brazil). The 5′UTR-ZIKV was amplified by PCR under the following conditions: 1 × of the enzyme buffer, 1.5 U of the Taq DNA polymerase enzyme, 1.5 mM MgCl2, 200 µM of each dNTP, and 5 µM of each primer (forward and reverse), brought to a final volume of 20 μL using ultrapure water. Thermocycling conditions were standardized to 30 cycles (95 °C–30 s, 60 °C–20 s, 72 °C–40 s) followed by 7 min of final extension at 72 °C. The amplified products were purified from a 1.5% agarose gel using 7.5 M ammonium acetate and ethanol for precipitation followed by centrifugation. Biotin was incorporated into the purified product by asymmetric PCR amplification, using biotinylated primer forward and dUTP (triphosphate deoxyuridine) instead of dTTP (triphosphate deoxythymine). The conditions of reagent concentration and thermocycling were the same as mentioned above. At the end of this reaction, RNA-like molecules were generated, that is, single-stranded DNA (ssDNA) molecules containing an uracil nitrogenous base, instead of thymine. The expected molecular weight of the amplified product was analyzed by agarose electrophoresis gel stained with ethidium bromide as reported elsewhere [80].

### 4.3. Library Construction and Ligands Selection

The RNA library used for this selection was composed of a pool obtained previously by our research group against non-coding regions of the DENV genome [21]. This pool contained ligands from the eighth round and non-ligands from the first round. This strategy was adopted due to reports of DENV and ZIKV co-infections [81,82]. Thus, by including the previously selected ligand pool, we could verify whether some of the sequences would be potential Zika ligands. The non-binding pool provided more significant variability of sequences discarded in the first round in the previous selection process, and we added it to select Zika-specific aptamers that could be useful in further studies.

To prepare the library, a pool was collected with the same concentration of these rounds, the initial library from the first pool contained products ranging from around 40 to 700 nucleotides, and the last varied from 41 to 187 nucleotides. The biotinylated target was incubated for 30 min at 37 °C with 80 µg Streptavidin MagneSphere^®^ Paramagnetic Particles (PMPs) previously washed in a saline sodium citrate solution (SSC). The non-immobilized and non-biotinylated strands were removed with 0.1 M NaOH. With the formation of the PMPs/target complex, 5 to 200 nmoles of a random RNA library pool was added into 500 µL of binding buffer at 37 °C for 1 h. Non-binding sequences were removed via four washes with the binding buffer (20 mM Tris pH 7.4, 100 mM NaCl, 5 mM KCI, 4 mM MgCl_2_, 3 mM CaCl_2_). Binding oligonucleotides were eluted twice by denaturation with 0.1 M NaOH at 37 °C, for 10 min. The oligonucleotides were precipitated with ethanol and then resuspended in water. Ligands were reverse transcribed and subsequently amplified by RT-PCR. At the start of the next round, a new pool of RNAs was generated by in vitro transcription using the T7 RiboMAX kit. Eight rounds of selection were carried out and the PCR product of the last round was sequenced to verify the variability of ligands and to analyze the sequences obtained.

### 4.4. Enrichment of the RNA Aptamers

Quantitative analysis of the enrichment of the oligonucleotide library over the selection was carried out by reading the RNA pool of eight each round in an L-Quant spectrophotometer (Loccus Biotecnologia, Cotia, Brazil) at wavelengths of 260 nm and 280 nm.

For the enrichment of the aptamer pools, 10 µM of purified product from the second, fourth, sixth, and eighth rounds of aptamers were applied separately in a positively charged nylon membrane. After drying at 80 °C for 10 min, the membrane was exposed to ultraviolet light. The membrane was hydrated with a hybridization buffer (AlkPhos Direct Labeling Reagent kit) for 15 min at room temperature. Next, 5 µM of the biotinylated 5′UTR-ZIKV target (ssDNA with T replaced by U, see above) was added. The samples were left to hybridize for 18 h at room temperature. 

The membrane was washed for 5 min two times, with washing buffer as indicated by the manufacturer, and then streptavidin—conjugated to alkaline phosphatase—was added for 30 min at room temperature. At the end of this step, the membrane was washed three times with a washing buffer for 10 min each. For colorimetric detection, the addition of alkaline phosphatase substrates, NBT/BCIP, at concentrations of 0.2 mg/mL of NBT and 0.4 mg/mL of BCIP were used. After 1 h, the test was completed, and the membrane was washed with water.

### 4.5. Next-Generation Sequencing and Data Analysis

The sample containing the aptamers selected in the eighth round was submitted to the next-generation sequencing (NGS) using the Illumina HiSeq 2500 platform service (GenOne, Rio de Janeiro, Brazil). As performed similarly by Zuker et al. [42], we conducted a preliminary analysis of the raw data of the sequences obtained and the homologous oligonucleotides were grouped into clusters, obtaining the representative sequences. The selection of ligands was based on the cluster with the highest frequency, supposedly indicative of the sequences with higher affinity for the 5′UTR-ZIKV. Representative sequences from the first two hundred clusters were aligned in the software Snapgene (from Insightful Science; available at snapgene.com, accessed on 4 April 2021), where they were compared with our previously selected anti-dengue aptamers [21]. This analysis was performed in order to identify anti-dengue and anti-zika putative specifics as well as bifunctional aptamers. Furthermore, we used a reverse capture assay of the ligands with PMPs and real-time PCR to evaluate the affinity and specificity of the ligand (Appendix A); and the aptamers selected against 5′-UTR from ZIKV were grouped according to base sequence similarity. The aptamer APTZC10 for Dengue and Zika virus was chosen for molecular detection in biological samples.

### 4.6. Chemical Modifications to Produce a Molecular Aptamer Beacon

In silico mutations, variable sequences were performed on the aptamer APTZC10 (GUAGUGGGGAUGGUGAUGAUGGGGAUGGUGGUGAUG) to produce a molecular aptamer beacon (MB), denominated APTAZC10-MB, due to insertions of five nucleotides at the 5′-end with CGCGG; and its complementary sequence was added at the 3′-end of the molecule. The secondary structure prediction of APTAZC10 was used for the molecular aptamer beacon designer, by predictions on the webserver Mfold [42] using monovalent salt at 1 M Na^+^ concentration and 25 °C temperature. The most stable secondary structure of APTAZC10 presenting the lowest free energy was chosen. Hybridization assays was performed with 4 µM of each aptamer, APTZC10 and APTAZC10-MB, as a probe to confirm the affinity with the target. Thus, samples used as targets were ssDNA and double-stranded DNA (dsDNA) from 5′UTR-DENV2, ssDNA, and dsDNA from 5′UTR-ZIKV. The complementary sequence of the aptamer was employed as a positive control, and PBS was used as a negative control. Hybridizations were carried out under the same conditions previously described in this study. 

To confirm biomolecular interactions, an aptamer-target titration was performed with APTAZC10-MB, modified at the 5′-end with 6-carboxyfluorescein (FAM) and with 6-carboxy-tetramethylrhodamine (TAMRA) at the 3′-end, incubated with 5′UTR-DENV2 and 5′UTR-ZIKV separately to monitor signal alterations based on the fluorescence resonance energy transfer (FRET) mechanism. The aptamer concentration was fixed at 50 nM and different concentrations of the target were used (0–200 nM) at 25 °C. Fluorescence measurements were conducted by scanning the ranges from 500 nm to 700 nm (FAM excitation, FRET Channel) and from 565 nm to 700 nm (Direct TAMRA excitation, Acceptor channel) on a fluorescence spectrometer (Cary Eclipse, Varian, Palo Alto, CA, USA). The wavelengths used for excitation and emission were 494 nm and 520 nm, respectively, for FAM and 555 nm and 580 nm, respectively, for TAMRA. The data obtained were analyzed using the software Origin 2018 (Origin Lab Corporation, Northampton, MA, USA). Binding was analyzed by changing the intensity of FAM and TAMRA while the dissociation constant (K_d_) was calculated using the software GraphPad Prism, Version 8 (GraphPad Software Inc., San Diego, CA, USA). Data were analyzed by non-linear regression, and the model of a specific binding site used the following equation to calculate K_d_:(1)$$Y=\frac{\left(B_{\max}\;X\right)}{\left(K_{d}+X\right)}$$
where Y is the specific binding of the target aptamer, B_max_ is the maximum specific binding; K_d_ is the equilibrium dissociation constant; X is the free concentration of the target. The effects of ionic strength and temperature on the conformation of the aptamer, labeled in its sequence (5′-FAM; 3′-TAMRA), were analyzed through changes in the intensities of fluorophores (Appendix A). The APTZC10-MB was also modified with biotin at the 5′-end of the aptamer for diagnosis purposes. The titration assay was performed in triplicate and the images (Figure 4A,B) display means with standard error. The dissociation constant (kd) was obtained through non-linear regression, and the model was a specific binding site. This method was chosen because it is recurrently used to calculate the kd [83,84].

### 4.7. APTA-RT-PCR for DENV and ZIKV Detections in Serum Samples

To evaluate the ability of the APTAZ10-MB to capture flaviviruses in serum followed by RT-PCR, we used samples of participants with and without Dengue symptoms. Laboratory diagnoses of the individuals were made with rapid immunochromatography tests to detect IgM and/or IgG and/or NS1. The protocol was approved by the Research Ethics Committee of the Federal University of Goiás, under CAAE number 59323816.9.0000.5083. Written consent was obtained from all individuals to use their samples. In total, 108 serum samples, collected from February to June 2019, were analyzed. After collection, the samples were processed between July and September 2019. In addition to serum samples, data of the participants such as sex, age, and blood count of the participants were compared with molecular data. To capture viral RNA from DENV or ZIVK or both, the bifunctional and biotinylated (bio) aptamer was used. The immobilization of the complex composed of biotin-APTAZ10-MB-viral RNA was joined to PMPs to separate the genetic material of the viruses by using a magnetic platform. The captured viral RNA was eluted in DEPC-treated water and subjected to RT using M-MLV enzyme at 37 °C for 1 h. The amplification of complementary DNA (cDNA) was performed by conventional PCR, under the conditions previously described in this study to amplify the target and specific set of primers for each flavivirus (Figure 6).

For DENV, the PCR was performed under the conditions described by Cnossen and collaborators [21]; and for ZIKV, the forward primer was 5′-TGAATCAGACTGCGACATGTCG-3′ and reverse was 5′-TGACCAGAAACTCTCGTTTCCA-3′, designed from the consensus sequence GTGAATCAGACTGCGACAGTTCGAGTTTGAAGCGAAAGCTAGCAACAGTATCAACAGGTTTTATTTTGGATTTGGAAACGAGAGTTTCTGGTCAT. Duplicates of the reactions to amplify 5′UTR-DENV2 and 5′UTR-ZIKV were made for each sample, and the products were subjected to the electrophoresis at 1.2% (*w*/*v*) agarose gel stained with ethidium bromide and were photographically documented.

### 4.8. Statistical Analysis

Statistical analysis was performed using the software GraphPad Prism, Version 8 (GraphPad Software Inc., San Diego, CA, USA). Correlation analyses by Fisher’s Exact Test were performed in BioEstat (Version 5.0), comparing the diagnoses obtained in APTA-RT-PCR with laboratory data (IgM, IgG, and NS1) and blood count data (platelets and white blood cells). Values of *p* < 0.05 were considered significant for all analyses.

## 5. Conclusions

The methodologies proposed herein led us to characterize the aptamers APTZC10 and APTZC10-MB, which due to their bifunctionality could act in the diagnosis of both flaviviruses, ZIKV and DENV, through a capture assay followed by RT-PCR. We believe this to be the first report that combines the use of RT-PCR together with aptamers to capture DENV and ZIKV in a complex biological matrix such as serum, without processing and RNA extraction, reducing not only the potential degradation of this molecule, but also time and costs. NGS was used as an important complementary tool to scan the immense list of ligands obtained at the end of the selection process for anti-zika aptamers. Modifications and/or conjugations of aptamers are suggested to increase the sensitivity and specificity of the methodology, which can help in the development of differential diagnosis and assist in the epidemiological surveillance. This platform can provide faster and cheaper detection techniques and enable studies to verify potential antiviral of the aptamer APTZC10-MB.

## Figures and Tables

**Figure 1 ijms-23-13866-f001:**
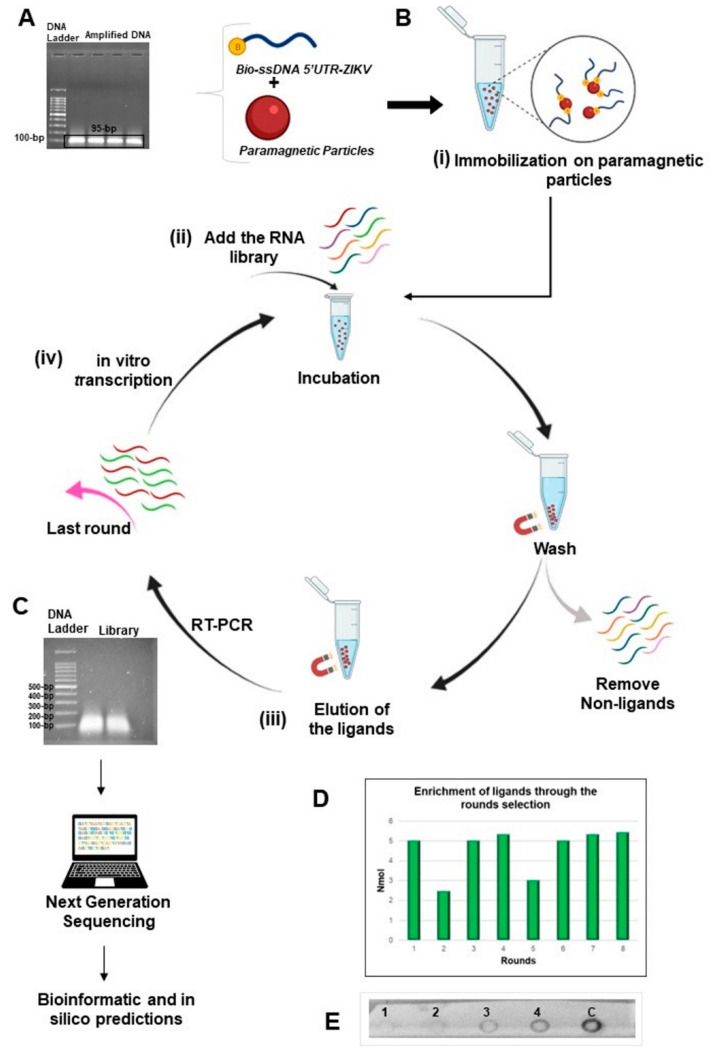
Representation of the method SELEX used in this study. (**A**) Electrophoresis agarose gel of the product of the ZIKV as target obtained from the 5′UTR synthetic sequence and presenting an expected size of 95-bp; (**B**) Steps for the selection process: (**i**) by using the biotinylated target immobilized on paramagnetic particles recovered with streptavidin; (**ii**) RNA library of ligands was added for incubation and the non-ligand oligonucleotides were removed; (**iii**) the binding molecules were eluted and amplified by RT-PCR; (**iv**) in vitro transcription of the amplified products initiated a new round for enrichment of RNA ligands; (**C**) Last round of ligand selection was analyzed by electrophoresis and then submitted to the next generation of sequencing (NGS). The oligonucleotides presented fragments with approximate size from 50 to 250-bp; (**D**) Enrichment of the ligands post-rounds of selection. The products of each round were quantified after in vitro transcriptions with a similar amount of the recovered products, except for the second and fifth rounds; (**E**) Hybridization by dot-blot assay demonstrating the enrichment and affinity of the ligands from the 2nd round (1st dot), 4th round (2nd dot), 6th round (3rd dot), 8th round (4th dot), and the dsDNA from 5′UTR-ZIKV as a positive control of the probe (5th dot). This figure was created through BioRender.com.

**Figure 2 ijms-23-13866-f002:**
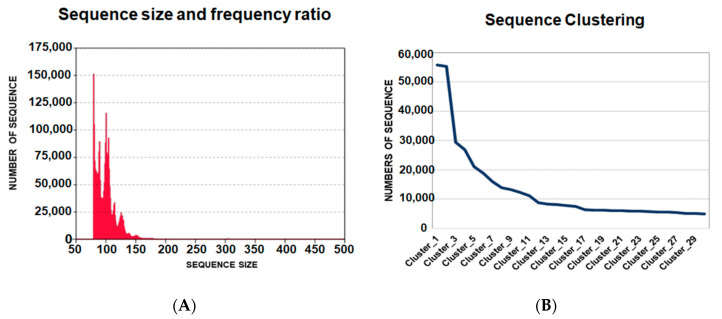
Data analysis of oligonucleotide ligands from next generation sequencing. (**A**) Frequency and size of the ligands of 5′UTR-ZIKV. The medium size of the ligands was 88-bp; (**B**) Clustering of the sequences from the highest to lowest frequencies of occurrence in the population of potential aptamers. The ligands were distributed from the first to the thirtieth clusters, where the first had more than 50,000 hits.

**Figure 3 ijms-23-13866-f003:**
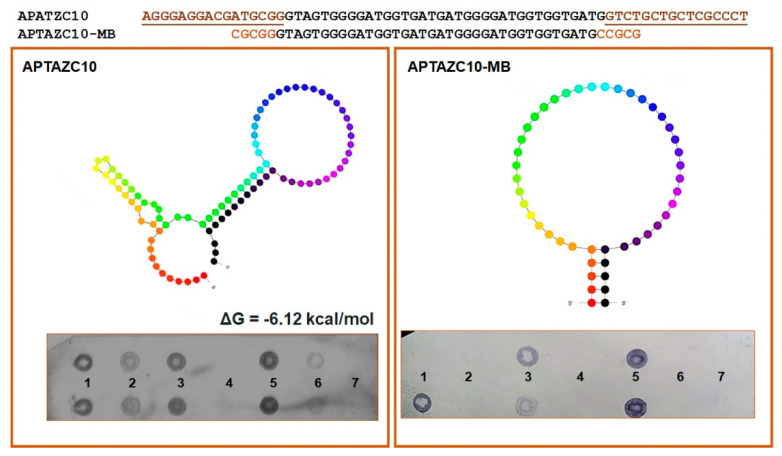
Characterization of the aptamers APTZC10 and APTZC10-MB. The APTZC10-MB was derived from APTZC10 as a molecular beacon, with a well-defined stem and loop structure. The dot-blot results allowed us to evaluate the affinity of both as ligands against: (1)—dsDNA 5′UTR-DENV; (2)—ssDNA 5′UTR-DENV2; (3)—dsDNA 5′UTR-ZIKV; (4)—ssDNA 5′UTR-ZIKV; (5)—dsDNA of the APTZC10 (positive control); (6)—ssDNA of the APTAZC10 (positive control); (7)—PBS buffer (negative control). The APTZC10-MB was more selective for the targets containing more than one strand in their structures.

**Figure 4 ijms-23-13866-f004:**
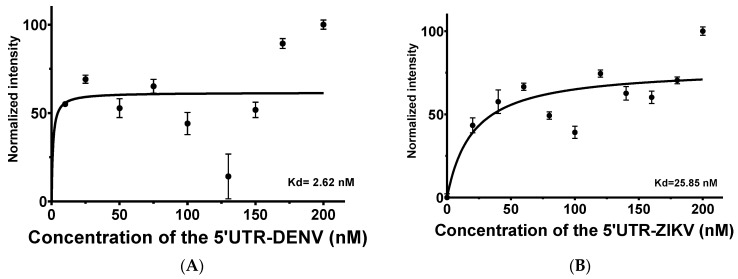
Aptamer-target titration in the presence of 50 nM of the APTAZC10-MB by FRET evaluations. (**A**) Dissociation constant of the APTZAC10-MB and 5′UTR of DENV2 interaction; (**B**) Dissociation constant of the APTZAC10-MB and 5′UTR-ZIKV interaction.

**Figure 5 ijms-23-13866-f005:**
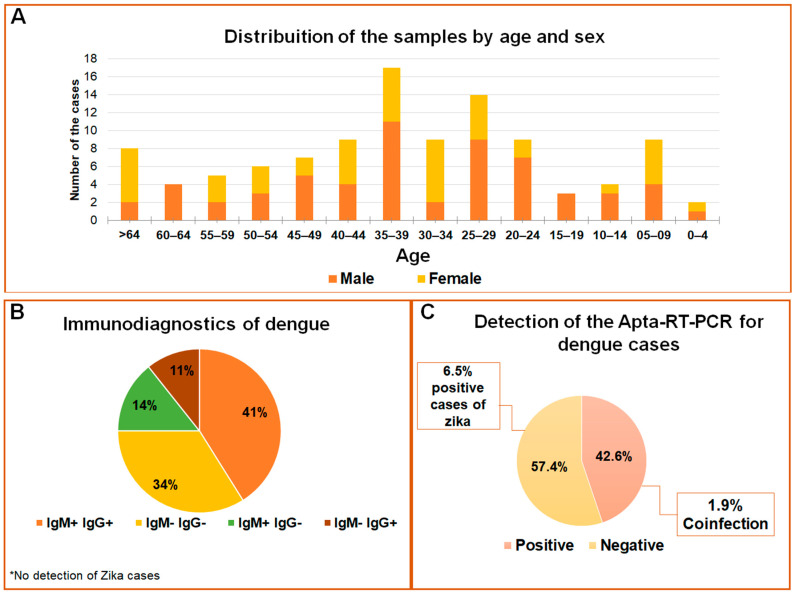
Profile of flavivirus detection in serum samples. (**A**) Population distribution according to age and sex of the individuals; (**B**) Distribution of the data according to IgM and IgG serological reactions; (**C**) Detections for DENV and ZIKV by APTA-RT-PCR in the studied population.

**Figure 6 ijms-23-13866-f006:**
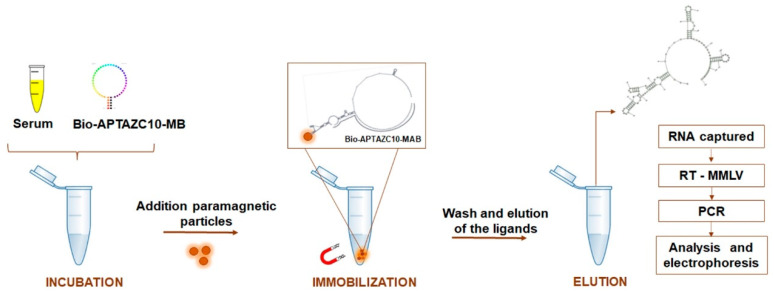
APTA-RT-PCR strategy for DENV and ZIKV detection. Biotinylated aptamer APTZC10-MB incubated directly with human serum, coupled with paramagnetic particles allowed the capture of viral RNA. After washes, the genetic material from the flaviviruses was submitted to RT-PCR, and the amplified products were analyzed by agarose electrophoresis.

**Table 1 ijms-23-13866-t001:** Comparison between methods used for diagnosis of flaviviruses.

Diagnostic Methods	Advantages	Disadvantages
VIRAL ISOLATION	- High sensitivity; - Viral serotype identification potential - Molecular characterization of samples [53]	- Blood samples collected early in the course of the disease - For Dengue cases: no differentiation between primary and secondary infections - Long culture incubation time that prevents early diagnosis - Specialized laboratory infrastructure for viral culture [54]
RT-PCR	- Rapidity of diagnosis - High sensitivity and high specificity [55]	- Possibility of cross-contamination between samples [56] - Several steps to perform the test
REAL TIME RT-PCR	- Reduction in time, test is done in a single step [56] - Higher sensitivity than viral isolation and conventional RT-PCR [57] - Low risk of contamination with the amplified product - Quantification of viral load [58]	- High cost [58]
PRNT	- Specificity for flaviviruses - Quantification of neutralizing antibody in serum samples [59]	- Difference in the interpretation of results - High cost - Complexity of the technique and long-time of realization - Difficulty in the differentiation of secondary and tertiary infections [60]
SEROLOGICAL TESTS	- Ease of use, specificity and sensitivity - Quantitative and qualitative detection of antibodies - Possible differentiation between primary and secondary infections [61]	- Cross-reactivity between flaviviruses [62]
RAPID DIAGNOSTIC TEST	- Ease of use - Speed of results - No specific laboratory equipment required [59]	- Lower sensitivity compared with ELISA-based assays - Performance differs between manufacturers [63] - Qualitative and operator-dependent visual result - Possibility of errors and inconsistency in the analysis

## Data Availability

Not applicable.

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
