# Peer review of "Development of a Molecular Aptamer Beacon Applied to Magnetic-Assisted RNA Extraction for Detection of Dengue and Zika Viruses Using Clinical Samples"

_ijms, 2022, doi:10.3390/ijms232213866_

Round 1

Reviewer 1 Report

The authors provided a method for detecting Dengue and Zika Virus based on an RNA aptamer-induced recognition. They first obtained an RNA aptamer that binds to the 5UTR of ZIKA. This aptamer can be further designed to probe the virus in clinical samples of patients with 295 ZIKV and DENV infections. The authors also compared the diagnostic accuracy of this method to the antibody-based methods, and believed that both ways are more suitable for clinical diagnostics with the advantages of low cost and fast data-turnaround time.  The manuscript can be strengthened by answering the following questions/addressing the concerns:

1.     Can the authors describe the information of the RNA library used in SELEX more clearly, and why they chose the pool which contained ligands from the eighth round and non-ligands from the first rounds (line382-383).

2.     The authors grouped the clusters of the sequences obtained from NGS, then selected the highest frequencies, cluster 1, to explore the representative sequences. Could the authors display the sequences of the top five cluster, I wonder if there is a similarity between these clusters.

3.     Table S1 showed that the sensitivity of APTA-RT-PCR ranging 217 from 35% to 60% and specificity between 52% to 63%. The diagnostic efficiency seems less than perfect, so how to improve the diagnostic efficiency of the method.

4.     The results expressed as a percentage would be easier to read and comprehend in Table S2.

5.     The authors calculated the Kd value via aptamer-target titration and the results were showed in Figure 4. However, the data of figure 4 was dispersion greatly, especially for Figure 4a. Did the authors repeated the assay for three times and obtain standard deviation. In addition, adding another method used to calculate KD value would be better.

6.     APTA-RT-PCR strategy were analyzed by agarose electrophoresis. However, it is hard to distinguish whether there were PCR amplified products in figure S5. Have the authors considered other methods, such as qPCR.

Author Response

  1. Can the authors describe the information of the RNA library used in SELEX more clearly, and why they chose the pool which contained ligands from the eighth round and non-ligands from the first rounds.

Answer: Due to reports of DENV and ZIKV co-infection [1,2], we aim to select aptamers that interacted in both the 5'UTR of ZIKV and DENV. Aptamers against DENV were previously and efficiently selected by our research group [3] using eight rounds of selection. So, based on our previous experience, we used a pool of ligands from the 8th round for ZIKV together with non-ligands for DENV as negative selection eluted in the first round, to obtain possible ZIKV-specific ligands. Also, to obtain aptamers able to interact with both viruses as targets, we used a pool of DENV-ligands from the 8th round. Our results demonstrated that this was an efficient strategy to select aptamers able to bind in the 5’UTR of ZIKV and DENV. We clarified this point in the “4.3. Library construction and ligands selection” (in blue).

  1. The authors grouped the clusters of the sequences obtained from NGS, then selected the highest frequencies, cluster 1, to explore the representative sequences. Could the authors display the sequences of the top five cluster, I wonder if there is a similarity between these clusters.

Answer: It was observed that in the sequence alignment, the first five clusters could be classified into three different groups, according to the linear motifs found: "ATGGG", "TGGGAT" and "AGGG. These motifs are variations of the "T/AGGG", which was observed more frequently in the sequences. This information was added in the Results, “2.2 Analysis of the sequences from Next-Generation Sequencing section” (in blue).

  1. Table S1 showed that the sensitivity of APTA-RT-PCR ranging 217 from 35% to 60% and specificity between 52% to 63%. The diagnostic efficiency seems less than perfect, so how to improve the diagnostic efficiency of the method.

Answer: We are aware of the limitations due to the lack of comparison of results with gold standard techniques such as real-time RT-PCR in the amplification step, to clearly assess the diagnostic efficiency of the technique presented. However, we analysed and compared it with epidemiological surveys that are carried out using serological techniques. In fact, it is observed that our protocol has advantages, as it uses aptamers in the RNA purification step, which reduces time and cost [4].  One of the disadvantages of the diagnostic technique used in this work is that viral RNA capture depends on the viremia level of the patients [5].  Thus, one way to improve efficiency would be to increase the affinity and specificity of the aptamer to capture RNA viral when it would be in small concentrations. For this, it would be necessary to look for other modification methods in the sequence and structure of the molecule [6]. As reported in the literature [7], another possibility to increase the affinity would be the conjugation of aptamers, since a sequence bank was generated, it would be interesting to analyze other sequences that can be conjugated with aptamers studied. Another strategy is to standardize another amplification reaction that amplifies virus coding sequences. Songjaeng et al. [8] reported that amplification of the 3'UTR-DENV region and a coding region increased the diagnostic sensitivity in samples collected during the viremia period. We included this comment in the Discussion section (in blue).

  1. The results expressed as a percentage would be easier to read and comprehend in Table S2.

Answer: We appreciate this comment and the results were expressed as a percentage in Table S2 (in blue).

  1. The authors calculated the Kd value via aptamer-target titration and the results were showed in Figure 4. However, the data of figure 4 was dispersion greatly, especially for Figure 4a. Did the authors repeated the assay for three times and obtain standard deviation. In addition, adding another method used to calculate KD value would be better.

Answer: The titration assay was performed in triplicate and the images (4a and 4b) were updated to display means with standard error. The dissociation constant (kd) was obtained through non-linear regression, and the model was a specific binding site. This method was chosen because it is recurrently used to calculate the kd [9,10]. We added this information in the “4.6. Chemical modifications to produce a molecular aptamer beacon” section (in blue).  Choosing the best method to calculate and obtain the best values requires independent and deep research. Lyu and colleagues [11], for example, developed an updated model to calculate kd focusing on cell-Selex. Thus, to add another calculation method in a non-superficial way, we would have to focus our research on updating the calculation methods reported in the literature.

  1. APTA-RT-PCR strategy were analyzed by agarose electrophoresis. However, it is hard to distinguish whether there were PCR amplified products in figure S5. Have the authors considered other methods, such as qPCR.

Answer:  Initially, the samples were amplified using qPCR using the Eva green kit (HOT FIREPol® Eva Green® qPCR Mix Plus ROX kit - Solis Biodyne). However, the reaction also favoured the amplification of primer dimers, resulting in false positive results. So, the qPCR could have an advantage if qPCR were performed with four different probes to amplify each of the different dengue serotypes, for example, which was not the objective of the work. Thus, we decided to carry out the amplification via conventional PCR analyzing the product by agarose gel, as reported elsewhere [12].

References

  1. Valiant, W.G.; Mattapallil, M.J.; Higgs, S.; Huang, Y.-J.S.; Vanlandingham, D.L.; Lewis, M.G.; Mattapallil, J.J. Simultaneous Coinfection of Macaques with Zika and Dengue Viruses Does not Enhance Acute Plasma Viremia but Leads to Activation of Monocyte Subsets and Biphasic Release of Pro-inflammatory Cytokines. Sci. Rep. 2019, 9, 7877, doi:10.1038/s41598-019-44323-y.
  2. Estofolete, C.F.; Terzian, A.C.B.; Colombo, T.E.; de Freitas Guimarães, G.; Ferraz, H.C.J.; da Silva, R.A.; Greque, G. V; Nogueira, M.L. Co-infection between Zika and different Dengue serotypes during DENV outbreak in Brazil. J. Infect. Public Health 2019, 12, 178–181, doi:10.1016/j.jiph.2018.09.007.
  3. Cnossen, E.J.; Silva, A.G.; Marangoni, K.; Arruda, R.A.; Souza, E.G.; Santos, F.A.; Fujimura, P.T.; Yokosawa, J.; Goulart, L.R.; Neves, A.F. Characterization of oligonucleotide aptamers targeting the 5′-UTR from dengue virus. Future Med. Chem. 2017, 9, doi:10.4155/fmc-2016-0233.
  4. Sukla, S.; Mondal, P.; Biswas, S.; Ghosh, S. A Rapid and Easy-to-Perform Method of Nucleic-Acid Based Dengue Virus Diagnosis Using Fluorescence-Based Molecular Beacons. Biosensors 2021, 11, doi:10.3390/bios11120479.
  5. De La Cruz Hernández, S.I.; Flores-Aguilar, H.; González-Mateos, S.; López-Martínez, I.; Ortiz-Navarrete, V.; Ludert, J.E.; Del Angel, R.M. Viral load in patients infected with dengue is modulated by the presence of anti-dengue IgM antibodies. J. Clin. Virol. 2013, 58, 258–261, doi:10.1016/J.JCV.2013.06.016.
  6. Hasegawa, H.; Savory, N.; Abe, K.; Ikebukuro, K. Methods for Improving Aptamer Binding Affinity. Molecules 2016, 21, 421, doi:10.3390/molecules21040421.
  7. Nomura, Y.; Yamazaki, K.; Amano, R.; Takada, K.; Nagata, T.; Kobayashi, N.; Tanaka, Y.; Fukunaga, J.; Katahira, M.; Kozu, T.; et al. Conjugation of two RNA aptamers improves binding affinity to AML1 Runt domain. J. Biochem. 2017, 162, 431–436, doi:10.1093/jb/mvx049.
  8. Songjaeng, A.; Thiemmeca, S.; Mairiang, D.; Punyadee, N.; Kongmanas, K.; Hansuealueang, P.; Tangthawornchaikul, N.; Duangchinda, T.; Mongkolsapaya, J.; Sriruksa, K.; et al. Development of a Singleplex Real-Time Reverse Transcriptase PCR Assay for Pan-Dengue Virus Detection and Quantification. Viruses 2022, 14, doi:10.3390/v14061271.
  9. Maradani, B.S.; Parameswaran, S.; Subramanian, K. Development and characterization of DNA aptamer against Retinoblastoma by Cell-SELEX. Sci. Rep. 2022, 12, 16178, doi:10.1038/s41598-022-20660-3.
  10. Siddiqui, S.; Yuan, J. Binding Characteristics Study of DNA based Aptamers for E. coli O157:H7. Molecules 2021, 26, doi:10.3390/molecules26010204.
  11. Lyu, Y.; Teng, I.-T.; Zhang, L.; Guo, Y.; Cai, R.; Zhang, X.; Qiu, L.; Tan, W. Comprehensive Regression Model for Dissociation Equilibria of Cell-Specific Aptamers. Anal. Chem. 2018, 90, 10487–10493, doi:10.1021/acs.analchem.8b02484.
  12. Ali, E.O.M.; Babalghith, A.O.; Bahathig, A.O.S.; Dafalla, O.M.; Al-Maghamsi, I.W.; Mustafa, N.E.A.G.; Al-Zahrani, A.A.A.; Al-Mahmoudi, S.M.Y.; Abdel-Latif, M.E. Detection of Dengue Virus From Aedes aegypti (Diptera, Culicidae) in Field-Caught Samples From Makkah Al-Mokarramah, Kingdom of Saudi Arabia, Using RT-PCR. Front. public Heal. 2022, 10, 850851, doi:10.3389/fpubh.2022.850851.

Reviewer 2 Report

This paper describes a technique for extraction of RNA of two important flaviviruses and subsequent detection of the RNA using a conventional RT-PCR with gel electrophoresis as the indicator system.

I am a little concerned that this paper describes a very interesting technique but there’s still a long way to go in evaluating the contribution that this technique can make to the diagnosis of dengue and Zika infections.

The indicator system as described in figure 6 is an RT-PCR that clearly has two primers and electrophoresis as the indicator system. Compared with TaqMan this indicator system is relatively insensitive and lacks specificity.

This technique needs to have a head-to-head comparison with RT-PCR using viral RNA extraction and TaqMan as the indicator compared with the technique that is being described. The fact that this has not been carried out should be mentioned in the discussion and it should be indicated that there is a long way to go before we can understand the contribution that this technique can make to the routine diagnosis of dengue and Zika virus infections.

There are standard techniques described by a range of sources including CDC using TaqMan RT-PCR for these viral infections. The authors should note that that this is the gold standard by which the new assay needs to be compared.

I would like to see a full comparison noting comments on relative sensitivity, specificity convenience and cost. Clearly this would not be possible in the present study. However, the authors need to note that until such studies are carried out the utility of this assay can only be speculated on.

The unique feature of this particular study is the way in which RNA is extracted from the samples. This extraction procedure includes magnetic beads from which the RNA is eluted. The convenience and cost of this extraction technique would appear to be relatively similar to conventional RNA extraction techniques that utilise magnetic beads. The study has noted some results from samples that may contain either dengue or Zika viruses. The comparison is with immunoassays. This is clearly not a valid comparison.

At least some comments need to be put into the discussion section noting the deficiencies in the present study.

Specific comments

Line 82

Sequences from different locations demonstrated variability in size. It is impossible for a sequence to demonstrate anything at all let alone differences in size. This implies that the sequence is actively participating in a process of demonstration. Clearly what is meant is that differences in size were observed and this correlated with sequences from different geographical locations.

Line 86

The genomes did not exhibit a changed confirmation. They are incapable of exhibiting anything. Changes in confirmation were detected. However, the sequences do not exhibit.

Lines 121 and 122

The membrane does not show staining. The staining is observed and there is no showing of great affinity. The affinity may have contributed to this. This paragraph needs to be reworded.

Line 175

I’m not sure what this line means the hybridisation presented the double-strand targets. Ensure that the sentence is explained.

Line 176

The single-strand DNA does not show less interaction. The interaction is observed or calculated. The DNA does not show this.

Line 192

This does not show changes .The changes were observed or calculated.

Supplementary material

Figure Legends figure S1.

The aptamers do not show CT. These figure Legends need to be rewritten noting that different CTs were observed. The aptamers do not show temperature of melting. This whole page needs to be rewritten to more accurately reflect the fact that observations were made and that the sequences et cetera were not responsible for demonstrating or showing changes.

Author Response

1- The indicator system as described in figure 6 is an RT-PCR that clearly has two primers and electrophoresis as the indicator system. Compared with TaqMan this indicator system is relatively insensitive and lacks specificity.

Answer: The authors agree with the reviewer that TaqMan system by using probes as one of the molecular strategies for genes and/or pathogen detections and it can be compared to conventional RT-PCR. On the other hand, the fact that qPCR was not used with probes in our work does not make molecular detection unfeasible, which is widely used in diagnostic detection studies around the world [1]. However, one of the disadvantages presented would be the need to design different probes for each serotype as targets and the assays would be more expensive for the research. Another lower-cost Eva Green reagent (HOT FIREPol® Eva Green® qPCR Mix Plus - Solis Biodyne) was used in the preliminary amplification assay. The fluorescent intercalating reagents in this kit can associate with double-stranded DNA and fluoresce brightly, however, this can occur with any double-stranded product, including primer dimers. The amplification conditions and the reagent used favored the formation of dimers in our results (data not shown), whose results could be confused with false positives.  As the conventional PCR reaction was optimized in our laboratory, we chose to carry out the analysis of the samples. Optimizations by using probes designed to discriminate the DENV viral serotypes in qPCR format would be carried out in future assays by our group.

2-This technique needs to have a head-to-head comparison with RT-PCR using viral RNA extraction and TaqMan as the indicator compared with the technique that is being described. The fact that this has not been carried out should be mentioned in the discussion and it should be indicated that there is a long way to go before we can understand the contribution that this technique can make to the routine diagnosis of dengue and Zika virus infections.

Answer: We appreciate this comment as an important issue for infectious disease and the possible use of qPCR and probes were mentioned in the Discussion section (in blue).

3-There are standard techniques described by a range of sources including CDC using TaqMan RT-PCR for these viral infections. The authors should note that that this is the gold standard by which the new assay needs to be compared.

Answer: The authors are aware of the limitations of the article due to the lack of comparison of the technique presented for diagnosis with gold standard techniques. As we agree with the reviewer, these limitations were included in the article in the Discussion section (in blue).

4- I would like to see a full comparison noting comments on relative sensitivity, specificity convenience and cost. Clearly this would not be possible in the present study. However, the authors need to note that until such studies are carried out the utility of this assay can only be speculated on.

Answer: The general aspects regarding to this point focusing evaluation and contributions of the techniques in routine exams were added in the article as a new table, the Table 1. The utility of how the technique described in our study could contribute to the diagnosis of dengue and zika viruses was included in the final part of the Discussion section (in blue).

5-The unique feature of this particular study is the way in which RNA is extracted from the samples. This extraction procedure includes magnetic beads from which the RNA is eluted. The convenience and cost of this extraction technique would appear to be relatively similar to conventional RNA extraction techniques that utilise magnetic beads. The study has noted some results from samples that may contain either dengue or Zika viruses. The comparison is with immunoassays. This is clearly not a valid comparison.

Answer: The authors are aware that there are magnetic bead kits on the market that can be used to extract RNA. As our aim was to combine the sensitivity of molecular RT-PCR techniques with the affinity of the aptamer, in the technique described, the beads act as a platform in the capture and the aptamers are the molecules that will bind and capture the target RNA. The aptamers will be selective in capturing and will avoid false results (positives or negatives).  The comparison with immunoassays was made because, despite not having a differential detection, this technique is routinely used due to its easy handling and low cost. RT-PCR techniques are recommended in epidemic surveillance studies, mainly post-mortem to confirm death cases due to dengue, for example. Nevertheless, the authors agree with the reviewer and this aspect was added as a complement in the final part of the Discussion section (in blue).

6-At least some comments need to be put into the discussion section noting the deficiencies in the present study.

Answer: We appreciate this comment and agree with the reviewer. The limitations of the study were mentioned in the Discussion section and the sentences mentioned in the specific comments have been rewritten (in blue).

Reference

  1. Sukla, S.; Mondal, P.; Biswas, S.; Ghosh, S. A Rapid and Easy-to-Perform Method of Nucleic-Acid Based Dengue Virus Diagnosis  Using Fluorescence-Based Molecular Beacons. Biosensors 2021, 11, doi:10.3390/bios11120479.

Round 2

Reviewer 1 Report

I think they have addressed the previous concerns. I have no further comments.